# Atomic Force Microscopy of Hydrolysed Polyacrylamide Adsorption onto Calcium Carbonate

**DOI:** 10.3390/polym15204037

**Published:** 2023-10-10

**Authors:** Jin Hau Lew, Omar K. Matar, Erich A. Müller, Paul F. Luckham, Adrielle Sousa Santos, Maung Maung Myo Thant

**Affiliations:** 1Department of Chemical Engineering, Imperial College London, London SW7 2AZ, UK; o.matar@imperial.ac.uk (O.K.M.); e.muller@imperial.ac.uk (E.A.M.); p.luckham01@imperial.ac.uk (P.F.L.); adryellesousasantos@gmail.com (A.S.S.); 2PETRONAS Research Sdn. Bhd., Lot 3288 & 3289, Off Jalan Ayer Itam, Kawasan Institusi Bangi, Kajang 43000, Selangor, Malaysia; maungmyothant@petronas.com

**Keywords:** atomic force microscopy (AFM), force spectroscopy, hydrolysed polyacrylamide, molecular weight, calcium carbonate

## Abstract

In this work, the interaction of hydrolysed polyacrylamide (HPAM) of two molecular weights (F3330, 11–13 MDa; F3530, 15–17 MDa) with calcium carbonate (CaCO_3_) was studied via atomic force microscopy (AFM). In the absence of polymers at 1.7 mM and 1 M NaCl, good agreement with DLVO theory was observed. At 1.7 mM NaCl, repulsive interaction during approach at approximately 20 nm and attractive adhesion of approximately 400 pN during retraction was measured, whilst, at 1 M NaCl, no repulsion during approach was found. Still, a significantly larger adhesion of approximately 1400 pN during retraction was observed. In the presence of polymers, results indicated that F3330 displayed higher average adhesion (450–625 pN) and interaction energy (43–145 aJ) with CaCO_3_ than F3530’s average adhesion (85–88 pN) and interaction energy (8.4–11 aJ). On the other hand, F3530 exerted a longer steric repulsion distance (70–100 nm) than F3330 (30–70 nm). This was likely due to the lower molecular weight. F3330 adopted a flatter configuration on the calcite surface, creating more anchor points with the surface in the form of train segments. The adhesion and interaction energy of both HPAM with CaCO_3_ can be decreased by increasing the salt concentration. At 3% NaCl, the average adhesion and interaction energy of F3330 was 72–120 pN and 5.6–17 aJ, respectively, while the average adhesion and interaction energy of F3530 was 11.4–48 pN and 0.3–2.98 aJ, respectively. The reduction of adhesion and interaction energy was likely due to the screening of the COO^−^ charged group of HPAM by salt cations, leading to a reduction of electrostatic attraction between the negatively charged HPAM and the positively charged CaCO_3_.

## 1. Introduction

Carbonate reservoirs house some 60% of global natural gas reserves, making them a source of lucrative business in the oil and gas industry [1,2]. However, most of these carbonate reservoirs have a relatively young geological age, whereby the rock grains do not experience sufficient natural cementation via deposition of minerals [3]; thus, these reservoirs are often weak and poorly consolidated [4]. Upon the extraction of hydrocarbon from the poorly consolidated reservoir, the increased effective stress imposed upon the formation rock grains is problematic as there is a possibility of abnormal effective stress increase that leads to irreversible deformation [5,6]. Rock grains are crushed under such deformation into fines, leading to reduced hydrocarbon production due to pore plugging, failure of processing equipment and increased cost of operation due to fines disposal [7]. Moreover, there is increased interest in using depleted reservoirs for carbon dioxide storage, and any disintegration of the reservoir is undesirable. In view of this issue, the oil and gas industry introduced primarily two mitigation strategies, namely mechanical and chemical techniques, to address fines production. The operational complexities of mechanical techniques, such as installation complexity, maintenance difficulties, reduced hydrocarbon production and possible wellbore damage [8,9,10,11] have steered the attention of the industry towards chemical techniques. 

The use of high molecular weight polymer has become an attractive alternative for the chemical consolidation of rock grains. An understanding of polymer interaction with rock surfaces is paramount to facilitating effective rock grain consolidation. Polymer retention mechanisms in reservoir rock are mainly via polymer adsorption, mechanical entrapment and hydrodynamic retention [12]. Mechanical entrapment and hydrodynamic retention are similar in that large polymer molecules are trapped either in the pore throats or the stagnant zones of porous rocks [13]. On the other hand, polymer adsorption involves the direct interaction between polymer molecules and a given surface. Adsorbed polymer molecules typically adopt configurations, as shown in Figure 1, where the attached segments are known as “trains”, and the free-floating segments form “loops” and “tails”, in which the tails give the effective hydrodynamic thickness (EHT) of the adsorbed polymer layers [14]. Adsorbed polymers could then consolidate rock particles via polymer bridging, where adsorbed polymer chains bridge particles and hold them together, or via charge neutralization, where charged polyelectrolyte electrostatically attracts rock particles together [15]. Polyacrylamide (PAM) is one such polyelectrolyte that has been employed extensively in many fields, including enhanced oil recovery (EOR), hydraulic fracturing, wastewater treatment and agricultural soil deconditioning [16,17]. 

Atomic force microscopy (AFM) can be employed to study polymer adsorption onto a given surface. AFM is a topographical imaging technique that involves the raster scans of the sample surface using a sharp tip [19]. As the AFM tip is brought into close proximity or even into contact with the sample surface, the interaction forces between the tip and the sample surface causes the cantilever to deflect. The cantilever deflection as the tip raster scans the surface produces topographical images. In a different mode, rather than scanning the tip over the surface, the tip is moved normally in relation to the surface. The deflection of the cantilever is monitored, from which the forces can then be calculated. The plot of surface interaction forces as a function of the surface separation is also known as force spectroscopy [20]. 

An AFM study on the adsorption of high molecular weight non-ionic polyacrylamide (NPAM) and cationic polyacrylamide (CPAM) on silica glass surfaces from an aqueous solution of different salinity (0.01 M and 0.34 M NaCl) was undertaken by Al-Hashmi et al. [18]. Force spectroscopy over several undisturbed areas [21] suggested low NPAM adsorption from 0.01 M NaCl on glass, in which the adsorption increased when NaCl concentration was increased to 0.34 M, due to the higher screening of the electrical double layers of the glass surface with a higher Na^+^ concentration. On the contrary, CPAMs in both solvents showed strong adsorption due to the strong electrostatic attraction between the two oppositely charged entities. They suggested that the strongly attracted CPAM molecules generally lay flat on the glass surface, while the weakly adsorbed NPAM molecules adopted a predominantly loops-and-tails configuration. Al-Hashmi et al., in another work [22], studied the adsorption of NPAM prepared in different salt solutions onto silica glass. Normally, NPAM primarily forms hydrogen bonding between the amide group of NPAM with the hydrogen atom of the silica surface. In a salt solution, the salt cations act as a salt bridge between the small amount of charged COO^−^ of NPAM and O^−^ groups of the silica surface. 

Grattoni et al. [23] conducted a similar study, where the AFM tip decorated with glass particles was brought to interact with a polyacrylamide (PAM)-incubated glass slide. When no salt was present, the glass surface remained unscreened, thus the slightly hydrolysed PAM induced very low surface coverage with a significant number of protruding loops and tails. As the loops and tails attached onto the AFM tip, the AFM tip retracted from the surface. Such an adhesive feature was lost when salt was present, as then the PAMs could lie flat on the screened glass surface. On the other hand, Li et al. [24] observed that, with increasing salt concentration, there was increased adhesion between SiO_2_ wafer adsorbed with PAM with an AFM tip incubated with polyethylene oxide (PEO)-coated silica dioxide (SiO_2_) colloidal probe. They argued that the salt cations screened the COO^−^-charged segment of PAM responsible for interacting with the positive hydroxyl group on the SiO_2_ wafer. These screened PAM segments could then adhere to and entangle with the approaching PEO-coated colloidal probe. 

In comparison with silica (or glass or silicon that all have surface hydroxyl (-OH) groups with which most pH’s become negatively charged), there have been few AFM studies on hydrolysed polyacrylamide (HPAM) interactions with calcium carbonate surfaces. One such study was undertaken by Ekanem et al. [13], where part of their study was on the adsorption of high molecular weight HPAM onto an Iceland spar calcite crystal. In their work, the calcite crystal was incubated in HPAM prepared in a 0.5% NaCl solution. A silica AFM tip was used for imaging and simple force measurement experiments. Results from their study showed that HPAM exhibited high surface coverage over the calcite surface, with regions of high, medium and low HPAM retention. They proposed that the high affinity between the charged carboxyl (COO^−^) group on HPAM polymeric segments and the positively charged calcite surface contributed to the high surface coverage of HPAM on the calcite surface [25,26]. For a region with higher HPAM retention, they suggested that such regions exhibited a higher height but lower Young’s modulus from imaging, and the repulsion distance was longer ranged, as was observed from the measurements, and vice versa was true. They proposed that regions with high HPAM concentrations contained large numbers of loops and tails, which contributed to higher heights, but were softer, as indicated by the lower Young’s modulus. The AFM tip also registered the repulsive forces exerted by the extruding tails, which was translated as a thicker polymer layer. 

In an earlier study of ours, we investigated the adsorption of hydrolysed polyacrylamide (HPAM) of different molecular weights onto calcium carbonate (CaCO_3_) particles [27]. To the best of our knowledge, there is not yet an AFM study that utilises an AFM tip with an attached CaCO_3_ particle to study the interaction between HPAM and a calcite surface. We believe this mode of study provides valuable insight on the intermolecular forces between HPAM and calcium carbonate, which could benefit efforts of carbonate reservoir consolidation using HPAM. Thus, in this paper, we study for the first time the effect of HPAM molecular weight and the effect of salt concentration on the interaction between HPAM and calcium carbonate.

## 2. Materials and Methods

### 2.1. Material

Iceland spar calcite crystals were purchased from Manchester Minerals (Manchester Minerals, Shrewsbury, UK). These crystals were colourless and rhombohedral. Two types of hydrolysed polyacrylamide (HPAM) were provided by SNF Floerger (SNF Ltd., Wakefield, UK), namely F3330S (30% hydrolysed, 11–13 MDa) and F3530S (30% hydrolysed, 15–17 MDa). The charged group of HPAM was due to the deprotonation of carboxyl groups of the acrylate monomer into the carboxylate group, as illustrated in Figure 2. The calcium carbonate powder (CaCO_3_, ≥99%) and sodium chloride (NaCl, 99.5%) were purchased from VWR Chemicals Ltd. (VWR Chemicals Ltd., Lutterworth, UK). All solutions used in this work were prepared using deionized water (DI, 18 MΩ). The AFM experiments were conducted under room conditions. 

### 2.2. Experimental Procedures

#### 2.2.1. Polymer Preparation

NaCl solution of 0.1 and 3 wt% were prepared by adding the measured salt into DI water stirred with a magnetic stirrer (Thermo Fisher Scientific, Dartford, UK). The 0.1% and 3% NaCl prepared corresponded to an ionic strength of approximately 0.017 M and 0.5 M, respectively. A low salt solution (0.1%) was prepared in order to remove any possible electrostatic interaction between bare surfaces that could interfere with the interpretation of the adsorbed polymer layers [28], while the high salt solvent (3%) was prepared to mimic naturally occurring reservoir conditions. The Debye–Hückel screening length, *κ*^−1^, calculated for 0.1% and 3% NaCl, were approximately 3 nm and 0.43 nm, respectively [18,29]. As for the polymer solutions, a 15,000 ppm polymer mother stock was first prepared according to API-RP-63, except that deionised water was used as the solvent instead of brine. In this methodology, deionised (DI) water was prepared in a beaker and set to a fast stirring speed of more than 400 rpm. The desired amount of polymer powder was then slowly poured around the shoulder of the vortex to prevent the formation of fisheye polymer clumps. The solution was stirred for at least 24 h for homogeneity. Once the mother stock was prepared, it was diluted into 100 ppm of stock solution using the 0.1 and 3 wt% NaCl solutions. 

#### 2.2.2. Calcite Crystal Incubation with HPAM

Iceland spar crystal was cleaved at room temperature in air with a clean-bladed tool to expose the original surface of the crystal. The cleaved sample surfaces were relatively smooth and typically possessed dimensions of approximately 10 mm × 10 mm × 3 mm and were thus suitable for nanoscale solid–liquid AFM experiments. The cleaved samples were immediately rinsed with DI water and absolute ethanol (Sigma Aldrich, ≥99.5%, Merck Life Science UK Limited, Dorset, UK) and were dried to remove any small calcium carbonate flakes from their surfaces. The cleaved samples were then placed in a clean petri dish, and the HPAM solution in 0.1 wt% NaCl was added slowly with a dropper until the crystals were fully immersed in the polymer solution. The petri dish was then immediately sealed with parafilm to prevent the intrusion of dust particles and incubated for at least 18 h, which, as our previous study showed, allowed sufficient time for complete polymer adsorption [27]. To study the effect of salinity, the HPAM in the 0.1 wt% NaCl solution was carefully extracted from the petri dish after the AFM experiments, and HPAM in 3 wt% NaCl was injected into the petri dish to fully immerse the calcite. Afterwards, the procedures were repeated.

#### 2.2.3. Atomic Force Microscopy (AFM)

In this work, a JPK NanoWizard 4 atomic force microscope was used to conduct the AFM experiments. The experiments were conducted using NanoWorld^®^ pyrex-nitride probe-triangular AFM cantilevers (PNPTR, NanoWorld® AG, Neuchâtel, Switzerland) (nominal force constant: 0.32 N/m; cantilever length: 100 µm; tip height: 3.5 µm; tip radius: <10 nm). Calcium carbonate particles (CaCO_3_, ≥99%, VWR Chemicals Ltd., Lutterworth, UK) with an average diameter of 3.5 µm were attached to the tip under a microscope using a micromanipulator (Melles Griot, Rochester, NY, USA). Since we could not procure single CaCO_3_ beads of the appropriate size, the CaCO_3_ powder we used was the same material used in our previous work [27], and, in our experimental protocol, instead of a single particle, a small cluster of particles was usually attached to the AFM tip. However, we did not believe this posed a concern, as it was most likely that a single CaCO_3_ particle would protrude from the cluster, making the final interaction with the calcite surface analogous to that of the single particle experiments in Al-Hashmi et al.’s or Grattoni et al.’s work [18,22,23,30]. In this study, the samples were scanned under force mapping (FM) mode (setpoint: 0.5 nN; vertical length: 300–500 nm; contact time: 2 s; pixels: 16 × 16–32 × 32), in which the interaction between the tip and the surface produced a complete force profile at each pixel [17]. All the force mapping experiments were conducted in liquid mode, where the AFM tip was immersed into the dilute polymer solutions and a scan was run on the calcite surface immersed in polymer solutions. The calibration of the AFM cantilever was conducted in liquid using contact mode. First, the AFM tip was made to contact a hard surface in the liquid, generating a force curve. The slope of the approach curve was selected, which told us the sensitivity of the cantilever with respect to the movement of the piezo. Then, the thermal noise measurement was conducted with a correction factor of 0.251. Hence, we obtained the cantilever’s stiffness, approximately 0.3 N/m, for all the cantilevers used. Note that, in all calculations, the actual force constant for each cantilever used was adopted in the force calculations. In each pixel, an approach and retract force curve was generated, in which the instrument collected a data point every 0.5 ms for each force curve. The approach velocity of the tip was set to 2 µm/s, as this speed gave us a reasonable experiment time without observing the hydrodynamic effect in the case of too high of an approach velocity [31]. In each experiment, a minimum of four FM scans were conducted, and the whole experiment was repeated at least twice using a freshly cleaved calcite surface and a new cantilever, meaning a minimum of 1000 force profiles were obtained. It is important to note that, since a fresh calcite surface and a new AFM tip modified with CaCO_3_ were used for each experiment, we anticipated a likelihood of the CaCO_3_ on the tip peeling away some of the HPAM molecules from the calcite surface during the first few pixels when the CaCO_3_ on the tip was relatively unoccupied. However, as the scans continued, the CaCO_3_ surface on the AFM tip was slowly saturated. We saw this as the system achieving equilibrium, and thereafter we observed the mutual detachment of HPAM molecules from both surfaces, that is, HPAM detached from the calcite surface onto CaCO_3_ on the AFM tip, and the HPAM detached from the CaCO_3_ on the tip onto the calcite surface. Here, the system stabilised, and we observed the net interaction of HPAM and CaCO_3_, regardless of the direction of detachment. Our primary objective was to compare the adhesion and interaction energy of each polymer considered in the present study with the calcite surface while observing details of the force profiles from force mapping, as the shape of the force profile could provide different insights into the polymer interactions with the calcite surface [32,33]. 

#### 2.2.4. Creep Rheological Measurement

Since long-chained polymer molecules are often coiled, it is rather difficult to calculate their end-to-end distance. A more helpful parameter in determining the size of a polymer molecule is the radius of gyration (R_g_), which is simply the average square distance from the center of the mass of the polymer coil to each polymer segment. Here, F3330 was selected to demonstrate the effect of salinity on the R_g_ of the polymer. R_g_ could be experimentally determined by conducting a creep rheological measurement. Here, a Thermo-Fisher Haake MARS 60 rotational rheometer (Thermo Scientific, Karlsruhe, Germany) was used to conduct the creep experiment. F3330 of various concentrations was prepared both in deionised (DI) water and in a 0.5 M salt solution. A double-gap cylindrical rotor, with an inner and outer diameter of 26 mm and 21 mm, respectively, and a length of 40 mm, was selected. The double-gap cylindrical rotor sat inside in a double-gap cup with an inner and outer diameter of 27 mm and 20 mm, respectively. A 3 mL of sample was required per measurement, and the rotor was immersed into the cup containing the sample until the base of the rotor was 4 mm from the base of the cup. Double-gap geometry was used as it is the most sensitive geometry in rotational rheometers. The F3330 samples in the double-gap cup were subjected to a low shear stress for 10 minutes, where a strain against the time plot could be obtained. The shear rate was then calculated from the linear region of the plot, and the zero shear viscosity at this very low shear rate was obtained by dividing the imposed shear stress with the calculated shear rate. The specific viscosity (*η_sp_*) can be calculated via Equation (1)
(1)$$\eta_{sp}=\frac{\eta_{0}}{\eta_{s}}-1\;$$
where $\eta_{0}$ is the experimental zero shear viscosity and $\eta_{s}$ is the viscosity of the solvent. $\eta_{sp}$ was then plotted against the polymer concentration, and the polymer concentration where a slope change was observed indicated the critical entanglement concentration (CEC), which was the concentration of the polymer where significant intermolecular overlapping and entanglement occurred [34,35]. This CEC was then used to calculate the R_g_ of the polymers in the DI water and in the 0.5 M salt solution.

#### 2.2.5. Zeta Potential Measurement

The interaction between hydrolysed polyacrylamide (HPAM) and calcium carbonate (CaCO_3_) in different salinity could also be better understood via zeta potential analysis. A charged particle is surrounded by a layer of counterions rigidly bound to the colloid and a diffuse layer where the ions are less firmly associated. The potential at the boundary between these two layers is the zeta potential [36,37,38]; it is a useful parameter to determine the nature of the colloid along with the colloidal system. When an electric field is applied on a charged particle suspending in an electrolyte, the particle will move towards the electrode of opposite charge while resisting the viscous forces from the liquid medium. Upon reaching an equilibrium, the zeta potential (*z*) of the charged particle can be calculated via Equation (2).
(2)$$z=\frac{U_{E}3\eta }{2\varepsilon f(\kappa \alpha )}$$
where *U*_*E*_ is the electrophoretic mobility, *η* is the viscosity of the medium and *ε* is the dielectric constant. *f*(*κα*) is known as the Henry’s function, where *κ* is the Debye length and *α* is the radius of the particle.

Here, an Anton Paar Litesizer 500 (Anton Paar, Ostfildern, Germany) was used for zeta potential measurement. F3330 at a concentration of 500 ppm was prepared in DI water, 0.1% NaCl (0.017 M) and 3% NaCl (0.5 M). A Smoluchowski approximation was selected, which is suitable for particles larger than 0.2 microns dispersed in a salt solution above 0.001 M [39]. Results obtained from the zeta potential analysis could be used to confirm the calculation of the polymer configuration from the creep rheological measurement.

## 3. Results

### 3.1. CaCO_3_–CaCO_3_ Interaction

Before performing AFM scans on polymer-laden calcite crystal, a control experiment of CaCO_3_–CaCO_3_ scan was conducted to investigate the bare-surface interaction. This experiment was conducted in two salt concentrations, namely 1.7 mM and 1 M NaCl, (0.1% and 3%, respectively). Figure 3 shows the force–distance curve of calcite incubated in 1.7 mM NaCl with the AFM cantilever attached with CaCO_3_ particles. 

During the approach, the repulsive interaction commenced at approximately 20 nm. This repulsive interaction continued until approximately 7 nm, where a slight attractive interaction due to van der Waals forces was observed, and the CaCO_3_ of the AFM tip jumped into contact with the calcite crystal. Such a “jump-in” phenomenon was also observed in bare glass–glass interactions in 0.01 M NaCl by Al-Hashmi [18,21], which fit very well the typical interaction curve as proposed by the DLVO theory [40,41]. On retraction, there was significant adhesion due to van der Waals forces between the CaCO_3_ particles on the AFM tip and the calcite surface, and this caused the particle to adhere with the calcite surface until the bending of the AFM cantilever produced enough force to detach the particle from the surface, with the force becoming zero at approximately 80 nm. Since the AFM tip most likely had multiple CaCO_3_ particles attached, the initial slight retraction at approximately 16 nm was likely due to the detachment of one of the CaCO_3_ clusters from the surface, while full detachment was observed at 80 nm. 

The force–distance curve at high salt concentration, namely 1 M NaCl, is shown in Figure 4. The force–distance profile of calcite crystal at 1 M NaCl was significantly different from that at 1.7 mM NaCl. By observing the approach curve, we saw that there was almost no repulsive interaction at 1 M NaCl, but a small attractive interaction of approximately 80 pN could be observed below 10 nm (see the inset in Figure 4). The magnitude of adhesion upon retraction in 1 M NaCl was also significantly larger than that in 1.7 mM NaCl, with a less pronounced “snap-off”. 

In the absence of a polyelectrolyte, the interaction between charged colloidal particles in an electrolyte is governed by the electric double layer (EDL) of counterions surrounding the colloidal particles [42,43,44,45,46]. This EDL causes repulsion between colloidal particles and maintains the stability of the colloidal system. The EDL Debye–Hückel screening length, $\kappa^{-1}$, for monovalent ions in an aqueous solution at 25 °C can be simplified to expression (3):(3)$$\kappa^{-1}=\frac{0.304}{\sqrt{I(M)}}\;nm$$
where *I* is the ionic strength of the solution in mol/L. Upon calculation, the screening lengths at 1.7 mM and 1 M NaCl were 7.4 nm and 0.3 nm, respectively. The repulsive interaction observed at approximately 20 nm from Figure 3 was close to the calculated Debye screening length, since we expected two layers of electrical double layer (EDL) (one from the calcite surface, another from the CaCO_3_ particles) on the AFM tip, which amounted to a total Debye length of approximately 15 nm. When the EDL of two similar charged particles overlapped as they approached one another, the interaction of their ionic clouds led to a repulsive force between the two particles [41]. When the salt concentration increased, the presence of electrolytes screened out the range of the counterions in the ionic cloud, which suppressed the effect of EDL between particles [47,48,49]. The screened EDL of CaCO_3_ in 1 M NaCl also explained why the adhesion force upon retraction was significantly higher than for 1.7 mM NaCl. At 1 M NaCl; the retraction force profile did not exhibit a clear detachment, as would be observed in other similar works [18,22,23,30,50]. We noted, though, that the AFM tip in our work was most likely attached to a cluster of CaCO_3_ particles instead of a single CaCO_3_ particle. Thus, during hard contact, as shown in Figure 4, it was likely that, while there was one dominant particle with the calcite surface, the neighbouring CaCO_3_ particles could impose a small degree of interaction with the surface. This explained the jerky nature of the retraction curve in Figure 4.

### 3.2. Effect of Polymer Molecular Weight

In this study, cleaved Iceland spar calcite crystals were separately incubated in two types of hydrolysed polyacrylamide (HPAM), namely F3330 (11–13 MDa, 30% hydrolysed) and F3530 (15–17 MDa, 30% hydrolysed) of concentration 100 ppm dissolved in 0.1 wt% NaCl. Unlike a model system of a single particle attached to the AFM cantilever, inspection of all the force profiles revealed differences, particularly on retraction of the surfaces; however, there were two dominant (common) types of force profiles obtained, which are presented in Figure 5 and Figure 6 for F3330 and 3530 in 0.1% NaCl, respectively. First, we prefaced that a net repulsive interaction was observed on approach and a net attraction interaction was observed on retraction. With this in mind, we saw, as shown in Figure 5 that, on approach of the surfaces, there was no interaction until the calcium carbonate surfaces were some 40–50 nm apart, whereupon a monotonically increasing force was observed. The difference really occurred on separation of the surfaces. Figure 5a (Type I) basically shows a strong adhesion 600 pN, which initially rapidly decreased and then became slightly jerkier until the surfaces were 300 nm apart. Figure 5b (Type II) shows a similar repulsion, but, on separation, there were two very distinct regions where the adhesion was observed: an initial adhesion of 400 pN, close to contact, and another at around 180 nm, with a minimum of 150 pN with zigzag behaviour in between.

Figure 6 demonstrates the higher molecular weight polymer F3530. Here, the repulsion was somewhat longer ranged, commenced at 70–80 nm, and again increased monotonically with decreasing surface separation. Figure 6a (Type III), on retraction, shows a strong adhesion at two points, contact and 75 nm, where the separation between the data points was indicative of the cantilever moving with no resistance to a lower attractive force; in between, there were numerous zigzag curves. Type III had a significantly smaller initial peak, followed by zigzag force curves, whereas Figure 6b (Type IV) only had zigzag force curves. 

The retraction data (i.e., the adhesion) were then analysed further by making histogram plots of the maximum adhesion force and the interaction energy, i.e., the integral adhesion distance data. Since the histogram plots in each independent experiment were qualitatively similar, we only presented the histogram plots from one of the experiments. However, we tabulated the average and peak adhesion and interaction energy values from every experiment. The histograms of adhesion and interaction energy are shown in Figure 7 and Figure 8, respectively, with the green curve showing that the histograms generally followed a normal distribution fit, while the average and peak results are tabulated in Table 1. (Note that the scales of the *X* axis in the two figures differ by an order of magnitude). 

Different studies on polyacrylamide (PAM) interactions and possible configurations based on the shape of retraction force curves have been carried out by several authors [32,33,51] when studying the detachment of a single PAM molecule from a silica surface, typically using a silicon nitride cantilever with a very sharp tip. Note that both silica and silicon nitride have surface hydroxyl (-OH) groups and so the nature of the two surfaces is very similar. This setup posed an operational difficulty in our work, as the HPAM used was likely to be more strongly attracted to the calcite crystal due to the opposite charges of the two surfaces. Thus, the HPAM was most likely to detach from the silicon nitride tip (being partially negatively charged) rather than the calcite surface. We could only measure HPAM detachment from CaCO_3_ by modifying the AFM cantilever by attaching CaCO_3_ particles to the cantilever. Since the diameter of the calcium carbonate particles (3.5 µm) was much larger than the radius of an AFM tip (10 nm), it was unlikely that we would observe single molecule detachment, but rather the detachment of a few molecules; however, this would be the relevant situation if one were considering the interaction between calcium carbonate particles in a calcium carbonate reservoir.

By observing the typical distance where repulsion force was first observed during the retraction of the AFM cantilever, we were able to compile the range of the steric repulsion distance of F3330 and F3530 in 0.1% NaCl. Referring to Table 1, we noticed that the distance of steric repulsion during the approach of the surfaces was smaller in the case of F3330 than in F3530. This was to be expected since F3330 had a lower molecular weight than F3530; thus the F3330 molecules could not protrude from the surface as much as the F3530 molecules. This observation also agreed well with the proposed theory by several authors [7,27,52], whereby low molecular weight polymers could adopt a flatter configuration on a surface than higher molecular weight polymers (see Figure 9). 

Regarding the data obtained from retraction, we noted that the absolute values of the adhesion and interaction energies were not reproducible. This did not represent the error in the experiment, simply the variation. Several factors could account for this: (a) the configuration of polymer molecules on the surface was different, both spatially and temporally; (b) there was a different number of HPAM molecules that came into contact with CaCO_3_ on the AFM cantilever during every pixel force profile and (c) there was a possible slight shift of CaCO_3_ particles from the CaCO_3_ cluster on the AFM cantilever. Although these factors were unavoidable in our experiments, it is clear from the histogram plots of Figure 7 and Figure 8, and the computed results in Table 1, that the magnitude of the adhesion and interaction energies in F3330 was significantly larger than in F3530. This shows that the polymer coating process onto the calcite crystal is reproducible. The interaction between HPAM and calcium carbonate has been reported to be primarily due to the electrostatic interaction between the charged COO^−^ group of the HPAM segments with the positively charged calcite surface [13,26,53,54], while to a lesser extent by hydrogen bonding of the amide group on HPAM with any negative ions on the calcite surface [1]. Since F3330 had a lower molecular weight than F3530, it could adopt a flatter configuration on the calcite surface, [7,27,52], thus having more contact with the surface than F3530, which resulted in a higher degree of electrostatic interaction and hydrogen bonding with the calcite surface than F3530. Thus, when the CaCO_3_ on the AFM tip adhered to the calcite crystal, the higher number of contact points was translated to a larger overall pulling force required to either detach the F3330 polymers from the calcite surface or to detach from the CaCO_3_ particles on the AFM tip from the F3330 polymer. In both cases, this would directly result in larger adhesion and interaction energies for F3330.

Figure 10 shows the traditional force profile of a single HPAM molecule lying relatively flat to the surface being pulled away from the surface using an AFM. Plateau-like retraction force curves are observed, as this reflects the train segments of the HPAM on a substrate surface, followed by a sharp drop as the AFM tip breaks away from the surface [51,55,56,57] (From a macroscopic viewpoint, this is analogous to the stick–slip behaviour when adhesive tape is pulled away from a surface). Similar behaviour is likely to be observed when many polymer molecules are involved, with some modification due to the different molecular weights and charge group distributions of each molecule and that some molecules may anchor the AFM tip at the same surface separation until the force applied is large enough to pull all the molecules off the surface. With this in mind, we depict in Figure 11a simple schematic illustration of likely mechanisms by looking at the case of Types I and II’s detachment of two F3330 molecules from the surface. Here, we considered F3330 in our illustrations, since, as discussed above, F3330 was more likely to assume a surface-hugging configuration than the higher-molecular weight F3530. 

From Figure 11, we assumed both Types I and II had the same initial polymer configurations on the surface with the same number of anchor points, as depicted in panel (a). Full detachment of the polymer molecules in both cases is depicted in (d). We proposed that the difference in detachment mechanisms between Types I and II was due to the number of anchor points detached in a single detachment event. If a large number of anchor points was detached simultaneously, then this would lead to a significant drop in the adhesion force value, as shown for Type I and (b) in Figure 11. In contrast, if there was a more even detachment in the number of anchor points, it was represented in the force–distance curve as a two-stage drop in the adhesion, as shown for Type II and (c) in Figure 11. The point of detachment of polymer molecules could be easily recognised by the significantly low-density data points spread on the slope of the retraction force curve (refer to the circled regions), in which the slope also signified the stiffness of the AFM cantilever (We note that the instrument collected one data point on the force curve every 0.5 ms). Our proposed illustrations for multiple polymer detachment is also very similar to that proposed by Long et al. [58], where, in multiple polymer detachment, they also observed multiple peaks similar to those shown in Type II in Figure 11.

Figure 12 displayed another retraction force curve pattern observed in our work comprised zigzag-shaped curves, in which the force became zero, or close to zero, then became attractive again; this scenario was most recurring in the adsorption of F3530 onto calcite in 0.1% NaCl, e.g., Figure 6b. Zhang et al. [51] observed a similar zigzagged pattern in the adsorption of associative polymer of HPAM and hexadecyl dimethyl allyl ammonium chloride onto a silica surface. They reasoned that such a pattern was due to the successive stretching of polymer chains and detachment of anchor points as the polymer molecule was pulled away from the surface. It is important to note that such stretching of the polymer chains is only possible if the latter are long and not lying relatively flat on the surface, which is expected for F3530 molecules on a calcite surface. Thus, we believe this explains the zigzag portion of the force curve in Figure 6, with the only difference being that, in our work, there were significantly more zigzags due to multiple F3530 molecule detachment events from the calcite surface. The plateau-like region observed between 0–100 nm in the Type III retraction force curve could also be related to the simultaneous detachment of multiple anchor points, as discussed above and illustrated in Figure 11b,c.

### 3.3. Effect of Salinity

Before conducting AFM scans on polymers prepared in higher salinity brine, we observed the effect of brine on the polymer configuration. The specific viscosity of F3330 in 0 M (DI water) and 0.5 M (3% NaCl) salt solutions was plotted, and the radius of gyration (R_g_) of F3330 in both conditions was calculated via the critical entanglement concentration (CEC) obtained. The specific viscosity plot and value of R_g_ are illustrated in Figure 13 and Table 2, respectively. 

From Figure 13, we observed that the specific viscosity of F3330 in 0 M salinity (DI) was higher than in 0.5 M, and thus the critical entanglement concentration (CEC) in 0 M was also lower than in 0.5 M. First, we needed to note that the CEC separated the concentration region into a semi-dilute unentangled region (below CEC) and a semi-dilute entangled region (above CEC). The plot of specific viscosity against concentration was plotted in a logarithmic scale, and the power law of the slope above and below CEC typically corresponded to the certain scaling law, depending on the presence of salt. The scaling law of semi-dilute polyelectrolytes in the presence or absence of salt is tabulated in Table 3 [59,60].

The data points obtained in the specific viscosity plot in Figure 13 fit followed a trendline (R^2^ usually > 0.96). The power of the trendline in each region with and without salt also corresponded with the scaling law obtained in Table 3, showing that the CEC obtained from Figure 13 did indeed divide the plot into semi-dilute unentangled and semi-dilute entangled regions. Upon calculating the radius of gyration (R_g_) of the two polymer systems, F3330 in 0 M had an R_g_ of approximately 200 nm, while that of F3330 in 0.5 M was only approximately 60 nm, as shown in Table 2. This was a clear indication of the screening of the charged carboxyl (COO^−^) group by the salt ions, while reducing the intramolecular repulsion between the charged segments. This meant that the F3330 molecules in 0.5 M were not as expanded as those in 0 M, thus showing a drop in R_g_ [18,24,61].

The change in the F3330 configuration in the presence of salt was further consolidated by the zeta potential result tabulated in Table 4.

Table 4 shows the zeta potential of 500 ppm F3330 in the brine solution of various ionic strengths; a clear trend was observed where, with the increase in ionic strength, the magnitude zeta potential of the polymer became smaller. First, we observed that the zeta potential values obtained were negative in value, which was expected, since the F3330 is an anionic polyacrylamide (PAM). In the absence of salt, the charged carboxyl group (COO^−^) exerted a great repulsive force, which caused the expansion of the polymer chain, increasing its viscosity and thus increasing the zeta potential, which could be clearly observed from Equation (2). With an introduction of salt into the system, the charged COO^−^ group was screened by the salt cations, causing a collapse in the polymer configuration. This led to a reduction in viscosity, as observed in Figure 13, which directly led to a reduction in magnitude of zeta potential, according to Equation (2). The extend of the COO^−^ group screening also increased with a higher salt cation concentration, evident of the smaller magnitude of zeta potential at an ionic strength of 0.5 M compared with at 0.017 M. 

With this in mind, we conducted the force mapping (FM) of hydrolysed polyacrylamide (HPAM), where 3% NaCl solution was used to replace 0.1% NaCl to prepare the HPAM solution. The histograms of the adhesion and interaction energies are shown in Figure 14 and Figure 15, with the normal distribution fit shown as the green curve, while the average and peak results are tabulated in Table 5. (Note again the different scales of the *X*-axis in the two figures.) The batch numbers in both Table 1 and Table 5 indicate that the experiments were conducted under the same setup. For example, the AFM tip and the crystal used for F3330 of batch 1 in Table 1 and Table 5 were the same. The only difference being in Table 1, where F3330 was prepared in 0.1% NaCl, and as the experiment finished, we carefully extracted the existing polymer solution and replaced it with F3330 in 3% NaCl, thus giving us the value of batch 1 in Table 5. Therefore, the comparison between Table 1 and Table 5 needs to be made between the same batch of the same polymer. First, we noticed the steric repulsion distance experienced during the extend stage for both F3330 and F3530 was higher for 3% NaCl than for 0.1% NaCl. When the concentration of salt rose, the increased amount of Na^+^ ions from the salt screened out the carboxyl (COO^−^) segment of HPAM more effectively, which caused a lower electrostatic attraction between HPAM and the positively charged calcite surface [18,24,61]. The result of this reduction in electrostatic attraction was that the HPAM molecules were more loosely bound to the surface, thus they could extend further and be detected by the AFM tip; hence, we observed a slight increase in the steric repulsion distance. Similar results have been observed for oppositely charged polyelectrolytes (poly-l-lysine) adsorbed to mica surfaces [62,63], where, in very low electrolyte solutions (10^−3^ mol/L), the polyelectrolyte lay flat on the surface but, in 10^−1^ mol/L, the polyelectrolyte extended away from the surface by around 50 nm. More loosely bound HPAM molecules also signified a significant decrease in both the adhesion and interaction energies for both polymers. The general values of F3530 in batch 1 were very low, as most of the force curves of this batch had very low adhesion. A possible reason may have been due to the amount of CaCO_3_ particles attached to the AFM tip being relatively smaller than those associated with the other batch. However, we were still able to obtain the same trend if we repeated the experiment from 3% NaCl down to 0.1% NaCl, albeit the magnitude of adhesion and interaction energy value differed. 

## 4. Conclusions

Atomic force microscopy (AFM) studies of Iceland spar calcite crystals incubated in hydrolysed polyacrylamide (HPAM) of two different molecular weights have been conducted. Force mapping using CaCO_3_ particles attached to a silicon nitride tip was used for this study. A control experiment in the form of force spectroscopy between CaCO_3_ of the AFM tip and a calcite surface at 1.7 mM and 1 M NaCl displayed force curves that agreed with the DLVO theory [40,41]. During the approach of the two surfaces at 1.7 mM NaCl (0.1%), repulsive interaction between CaCO_3_ of the AFM tip with the calcite surface was observed, and the distance where a repulsive force was first recorded correlated well with the calculated Debye–Hückel screening length. As the salt concentration was increased to 1 M NaCl (3%), no repulsive interaction was observed due to the screening of the electrical double layer by salt ions. Adhesion due to van der Waal forces was observed during retraction in both salt concentrations, where the magnitude of adhesion was larger for 1 M NaCl than for 1.7 mM NaCl. A force mapping study of calcite crystal incubated in the HPAM F3330 and F3530 at 0.1% NaCl showed that the steric repulsion distance of F3530 was longer ranged than that of F3330. This was due to F3530 molecules being higher in molecular weight and consequently being able to extend further to be detected by the approaching AFM tip. On the other hand, greater adhesion and interaction energies were observed in F3330 than in F3530, suggesting that the lower molecular weight polymers adopted a flatter configuration on the substrate surface with more train segments, thus increasing polymer–surface interactions. When the salt concentration was increased from 0.1% NaCl to 3% NaCl, the adhesion and interaction energies of HPAMs with a calcite surface decreased due to the reduced electrostatic attraction between the charged groups on the polymer and the surface. The results obtained from this work would be instrumental in providing the groundwork for subsequent macroscale experiments, namely the consolidation of the calcium carbonate (CaCO_3_) plug using polyacrylamide (PAM), and investigating the geomechanical strength of the consolidated calcium carbonate samples in the form of unconfined compressive stress (UCS) analysis. 

## Figures and Tables

**Figure 1 polymers-15-04037-f001:**
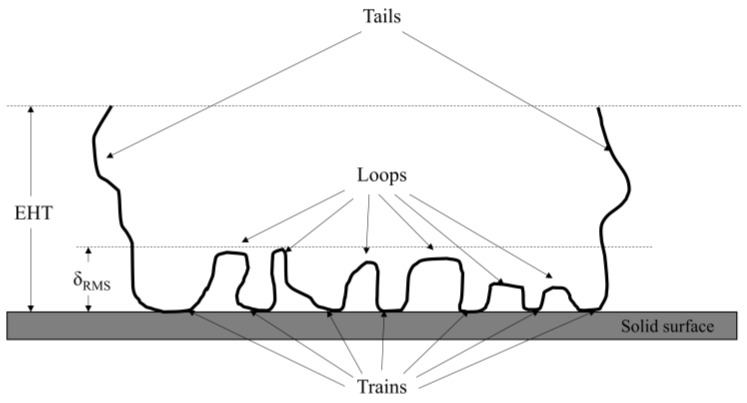
Schematics of polymer adsorption model with attached segments (trains) separated by unattached segments (tails and loops), adapted from Al-Hashmi et al. [18].

**Figure 2 polymers-15-04037-f002:**
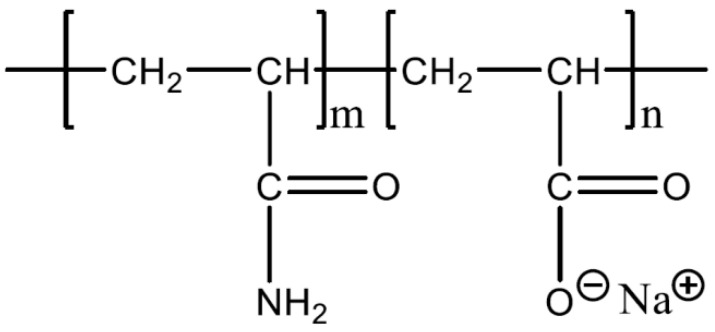
Molecular formula of HPAM.

**Figure 3 polymers-15-04037-f003:**
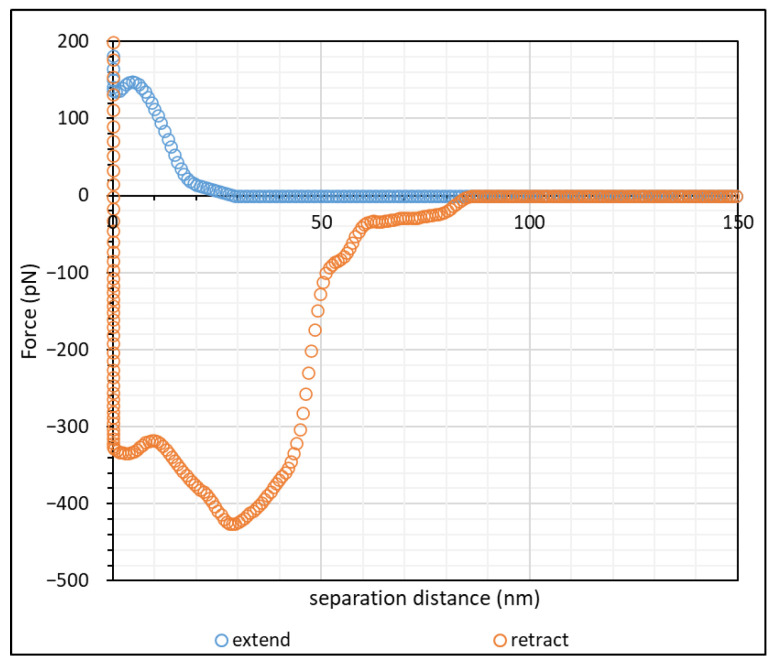
Force–distance curve of calcite crystal incubated in 1.7 mM NaCl solution.

**Figure 4 polymers-15-04037-f004:**
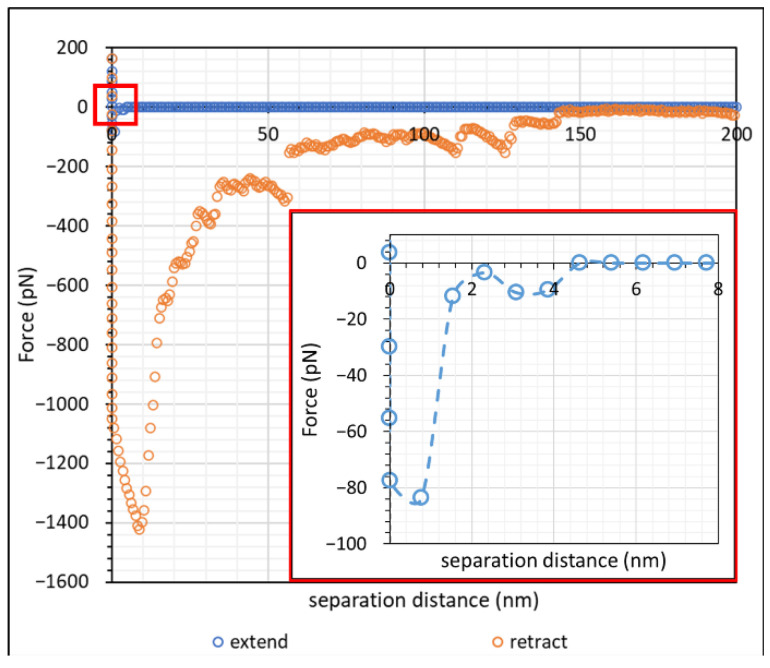
Force–distance curve of calcite crystal incubated in 1 M NaCl solution. Inset: extend force distance curve in red-squared region.

**Figure 5 polymers-15-04037-f005:**
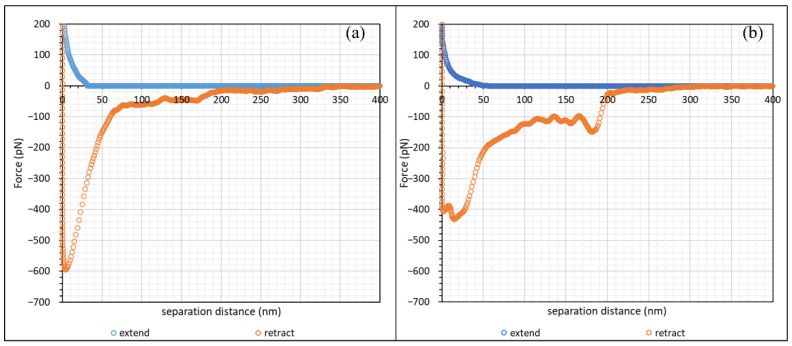
The two commonest forms of retraction profiles observed for F3330 in 0.1% NaCl. (**a**) Type I; (**b**) Type II.

**Figure 6 polymers-15-04037-f006:**
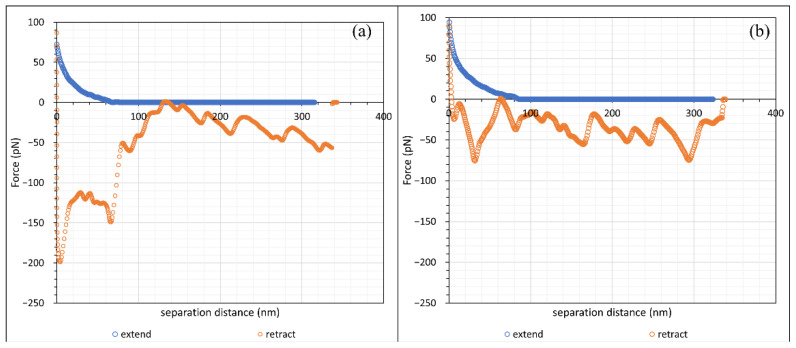
The two commonest forms of retraction force curve shape at F3530 in 0.1% NaCl. (**a**) Type III; (**b**) Type IV.

**Figure 7 polymers-15-04037-f007:**
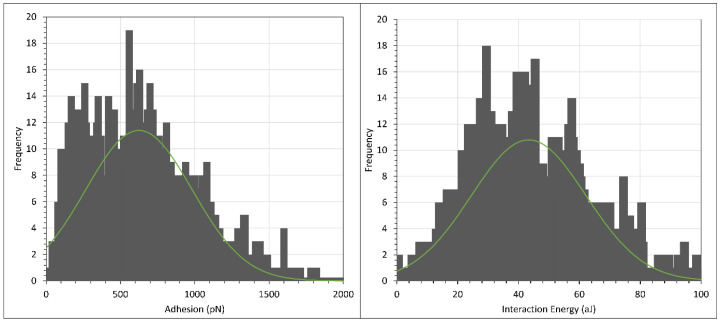
Histogram plot of adhesion (**left**) and interaction energy (**right**) of calcite immersed in F3330S in 0.1% NaCl. Green curve shows normal distribution plot.

**Figure 8 polymers-15-04037-f008:**
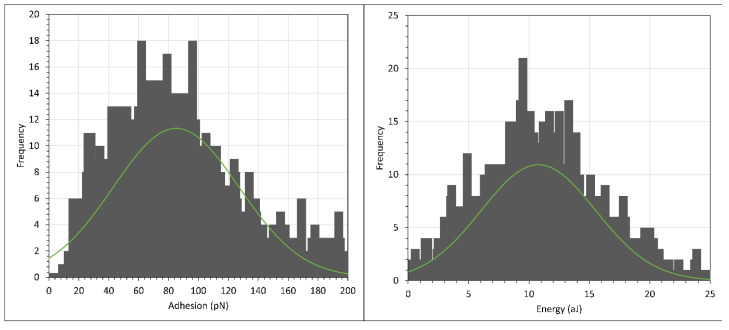
Histogram plot of adhesion (**left**) and interaction energy (**right**) of calcite immersed in F3530S in 0.1% NaCl. Green curve shows normal distribution plot.

**Figure 9 polymers-15-04037-f009:**
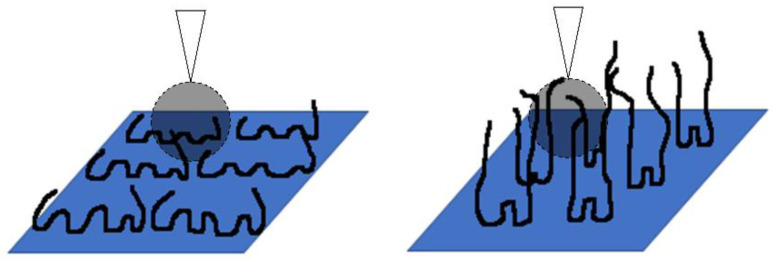
Cartoon representation of the possible configurations adopted by F3330 (**left**) and F3530 (**right**) on the calcite surface and their interactions with a CaCO_3_ particle attached to an AFM tip.

**Figure 10 polymers-15-04037-f010:**
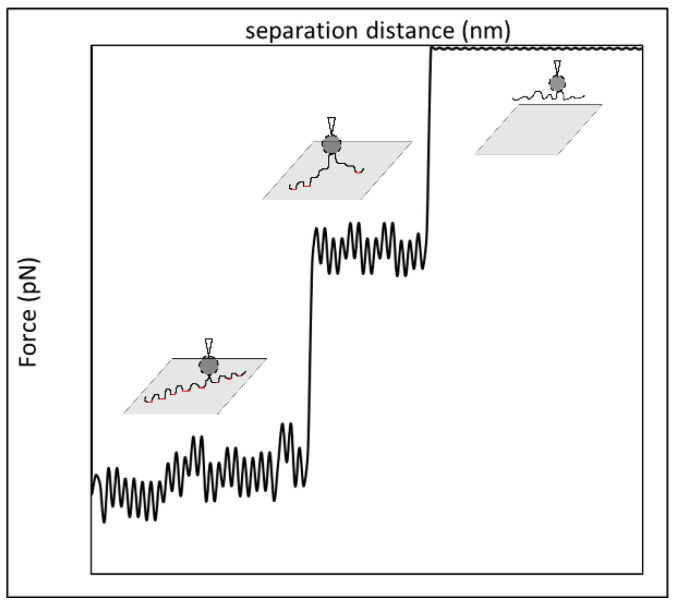
Force–distance curve of a single HPAM molecule (black) with probable HPAM detachment mechanisms from the substrate surface, adapted from Zhang et al. [51]. Anchor points between polymers with the surface are emphasized in red.

**Figure 11 polymers-15-04037-f011:**
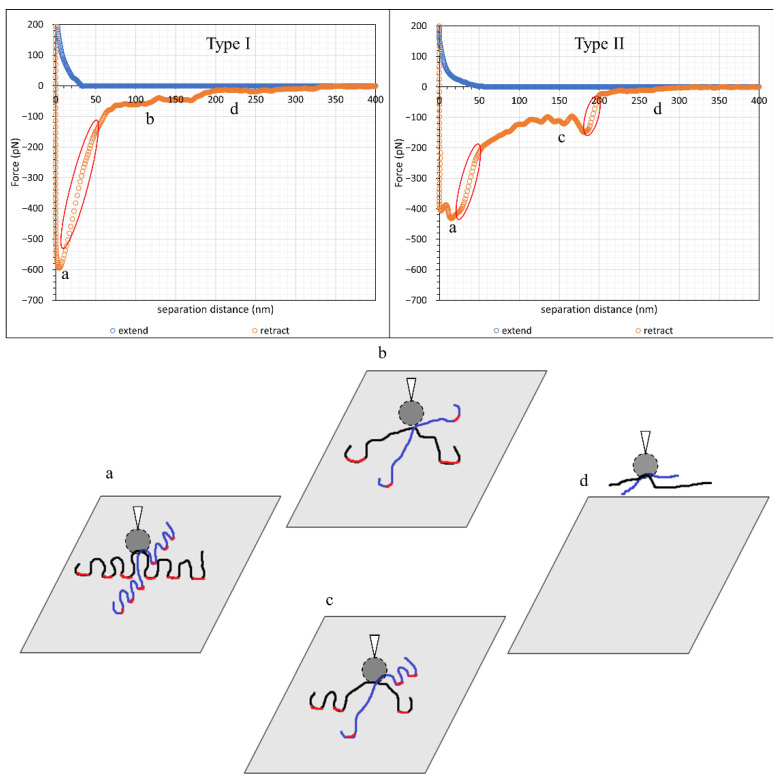
Force–distance curve for multiple detachment from more than one F3330 molecule (**top**) with proposed illustrations (**bottom**). Here two polymer molecules are coloured differently (blue and black) for the sake of clarity. Anchor points are emphasized in red. Different stages of polymer detachment are as followed: (**a**) Contact of CaCO_3_ particles with both polymer molecules at rest; (**b**) detachment of majority of the anchor points of polymer molecules from the surface; (**c**) even detachment of anchor points of polymer molecules from the surface; (**d**) full detachment of polymer molecules from the surface.

**Figure 12 polymers-15-04037-f012:**
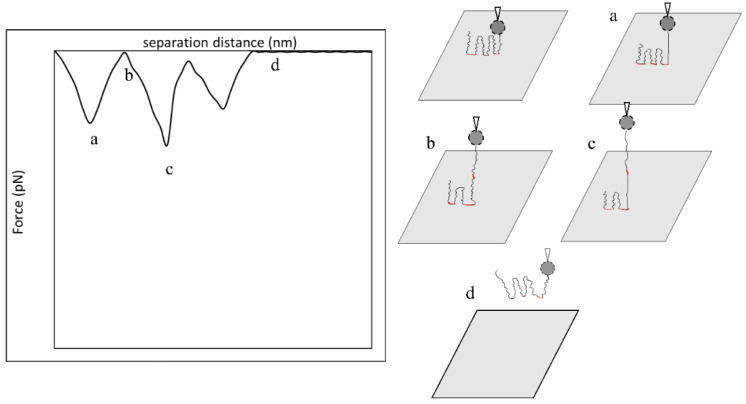
Illustration of possible single HPAM molecule (black) detachment that forms zigzag retraction force curve. Adapted from Zhang et al. [51]. Anchor points are emphasized in red. Different stages of polymer detachment are as followed: (**a**) tension due to detachment of polymer’s first anchor point from the surface produced the first adhesion peak; (**b**) stretching of the polymer molecule before second anchor point with no tension observed; (**c**) tension due to detachment of polymer’s third anchor point from the surface produced the third adhesion peak; (**d**) complete detachment of polymer molecule from the surface.

**Figure 13 polymers-15-04037-f013:**
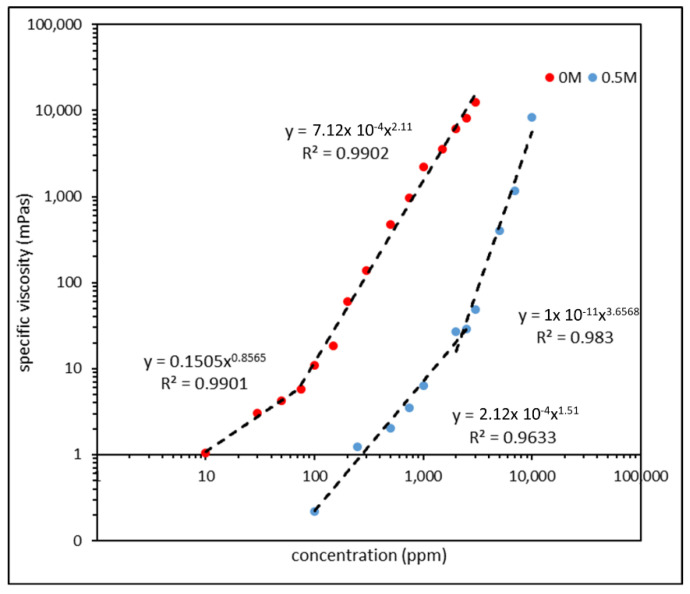
Specific viscosity plot of F3330 in 0 M (DI water) and 0.5 M salinity. CEC of F3330 in 0 M and 0.5 M are approximately 75 ppm and 2250 ppm, respectively.

**Figure 14 polymers-15-04037-f014:**
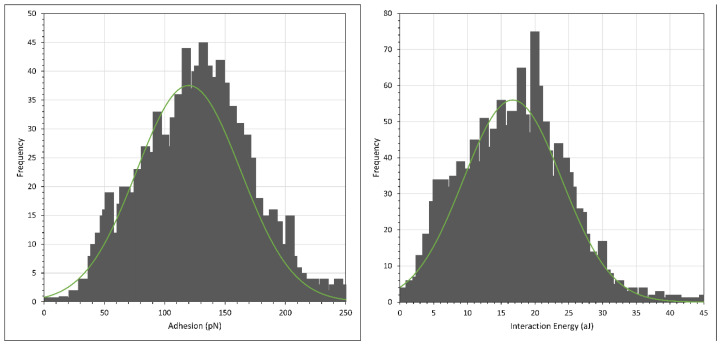
Histogram plot of adhesion (**left**) and interaction energy (**right**) of calcite immersed in F3330 in 3% NaCl. Green curve shows normal distribution plot.

**Figure 15 polymers-15-04037-f015:**
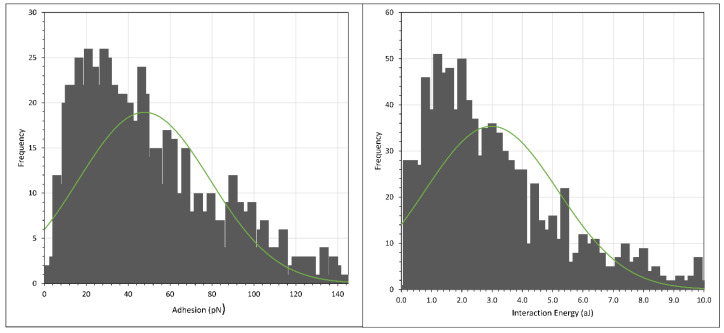
Histogram plot of adhesion (**left**) and interaction energy (**right**) of calcite immersed in F3530 in 3% NaCl. Green curve shows normal distribution plot.

**Table 1 polymers-15-04037-t001:** F3330 and F3530 steric repulsion distance, adhesion and interaction energies with calcite crystal in 0.1% NaCl.

Polymers	Steric Repulsion Distance (nm)	Batch	Adhesion (pN)		Interaction Energy (aJ)	
			Average	Peak	Average	Peak
F3330	30–70	1	625 ± 360	556 ± 50	43 ± 19	29 ± 8
		2	450 ± 320	266 ± 50	145 ± 88	66 ± 20
F3530	70–100	1	85 ± 42	61 ± 6	11 ± 4.7	9.5 ± 2
		2	88 ± 50	53 ± 4	8.4 ± 6.4	2.6 ± 2

**Table 2 polymers-15-04037-t002:** Radius of gyration (R_g_) of F3330 in 0 M and 0.5 M salinity.

Ionic Strength (M)	Radius of Gyration (nm)
0	200 ± 2.2
0.5	64 ± 2.3

**Table 3 polymers-15-04037-t003:** Scaling law for semi-dilute polyelectrolyte with and without salt.

Presence of Salt	Semi-Dilute Unentangled	Semi-Dilute Entangled
No	$\eta_{sp}\;\approx \;c^{0.5}$	$\eta_{sp}\;\approx \;c^{1.5}$
Yes	$\eta_{sp}\;\approx \;c^{1.25}$	$\eta_{sp}\;\approx \;c^{3.75}$

**Table 4 polymers-15-04037-t004:** Zeta potential of 500 ppm F3330 in DI, 0.017 M (0.1% NaCl) and 0.5 M (3% NaCl) salt solution.

Ionic Strength (M)	Zeta Potential (mV)
0	−41.1 ± 1.62
0.017	−25.3 ± 0.34
0.5	−14.7 ± 1.84

**Table 5 polymers-15-04037-t005:** F3330 and F3530 interaction with calcite crystal at 3% NaCl.

Polymers	Steric Repulsion Distance (nm)	Batch	Adhesion (pN)		Interaction Energy (aJ)	
			Average	Peak	Average	Peak
F3330	60–120	1	120 ± 43	131 ± 20	17 ± 7.3	23 ± 2
		2	72 ± 46	32 ± 5	5.6 ± 3.6	2.9 ± 0.25
F3530	90–150	1	11.4 ± 8.9	2.5 ± 2	0.3 ± 0.27	0.02 ± 0.01
		2	48 ± 31	18 ± 8	2.98 ± 2.19	1.6 ± 0.4

## Data Availability

Not applicable.

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
