# Peer review of "Atomic Force Microscopy of Hydrolysed Polyacrylamide Adsorption onto Calcium Carbonate"

_polymers, 2023, doi:10.3390/polym15204037_

Round 1

Reviewer 1 Report

J.H. Lew et al investigated the impact of HPAM molecular weight and salt concentration on the interaction between HPAM and calcium carbonate. Extensive work has been done to study and characterize this system using the AFM technique. The general subject of the manuscript is interesting; however, several questions arise that should be addressed before publication.

1.      Whether the coating process (adsorption of polymer on CaCO3) was reproducible?

2.      „Adsorbed polymer molecules typically adopt configurations as shown in Figure 1, where the attached segments are known as ‘trains’, and the free-floating segments form ‘loops’ and ‘tails’, in which the tails” The figure is very illustrative and can be misleading. Polyelectrolytes can adopt highly variable conformations depending on the external environment (ionic strength, pH, type of electrolyte). Did Authors perform any additional experiments leading to unraveling the PAM conformations?

3.      Did the authors perform any physicochemical characteristics of PAM in electrolyte solutions? Data about zeta potential dependence on ionic strength would be beneficial in analyzing electrostatic interactions with CaCO3.

In addition, the information on the practical applications of the obtained results and vision for future work should be extended. How to transfer knowledge obtained in the laboratory to a macro scale?

Author Response

We wish to thank the reviewer for having read our manuscript and for the positive comments and suggestions to improve it. We know how much time it takes to review manuscripts and we sincerely appreciate the observations. We have acted on all of the remarks, and we feel they have improved significantly the manuscript.

Below we make a point-by-point discussion. The original reviewer’s comments are placed in italic black the response is red below each point.

Reviewer 1

  1. Whether the coating process (adsorption of polymer on CaCO3) was reproducible?

Author reply:

The coating process of the polymer onto the calcite crystal was a rather straightforward process, and therefore it is reproducible. This could be observed from the trend of the results. Force mapping results from F3330 is always higher than that of F3530. However, we do mention in the manuscript that the absolute magnitude of the adhesion and interaction energies are not always reproducible due to a) the configuration of polymer molecules on the surface is different, both spatially and temporally; b) the different number of HPAM molecules that came into contact with CaCO3 on the AFM cantilever during every pixel force profile and c) a possible slight shift of CaCO3 particles from the CaCO3 cluster on the AFM cantilever. These are factors out of our control, but we believe our sample sizes (scan number > 1000) are large enough that a consistent trend could be observed.

Full explanation is in line 411 – 421.

  1. Adsorbed polymer molecules typically adopt configurations as shown in Figure 1, where the attached segments are known as ‘trains’, and the free-floating segments form ‘loops’ and ‘tails’, in which the tails” The figure is very illustrative and can be misleading. Polyelectrolytes can adopt highly variable conformations depending on the external environment (ionic strength, pH, type of electrolyte). Did Authors perform any additional experiments leading to unraveling the PAM conformations?

Author reply:

The polymer configuration was estimated in the form of radius of gyration (Rg) through a creep rheological analysis. The full explanation of the experimental setup for this analysis is explained in section 2.2.4. From the creep rheological analysis, we could obtain the critical entanglement concentration (CEC) which informs us the concentration where significant intermolecular overlapping and entanglement occurred. The value of CEC could be used to calculate the Rg of the polymer in the presence or absence of brine (0M vs 0.5M).

Results from creep rheology (line 494 - 524) revealed that the Rg of the PAM decreased in the presence of salt, which shows a change in polymer configuration from expanded to a more coiled structure. This is a direct consequence of the screening of the COO- charged group on the PAM segments, which reduces intramolecular repulsion responsible of expanding the polymer chains.  

Ionic strength (M)

Radius of gyration (nm)

0

200 ± 2.2

0.5

64 ± 2.3

  1. Did the authors perform any physicochemical characteristics of PAM in electrolyte solutions? Data about zeta potential dependence on ionic strength would be beneficial in analyzing electrostatic interactions with CaCO3.

Author reply:

Zeta potential analysis of PAM in DI water, salt solutions of 0.017M and 0.5M were conducted. The experimental procedures are explained in section 2.2.5.

Results from zeta potential analysis (line 525 - 542) shows that the magnitude of zeta potential of PAM decreases as the ionic strength of the solvent increased. This supports the result obtained from creep rheological analysis (line 494 - 524), where we expect there is a screening of the charged COO- group on the PAM chain by the introduction of salt cations. The extend of COO- group screening also increases with higher salt cation concentration, evident of the smaller magnitude of zeta potential at ionic strength of 0.5M compared to that at 0.017M. We also observed that the zeta potential values obtained were negative in values, which is expected since the PAM used in this work is an anionic polyacrylamide (PAM).

Ionic strength (M)

Zeta potential (mV)

0

-41.1 ± 1.62

0.017

-25.3 ± 0.34

0.5

-14.7 ± 1.84

  1. In addition, the information on the practical applications of the obtained results and vision for future work should be extended.How to transfer knowledge obtained in the laboratory to a macro scale?

Author reply:

The results obtained from this work would be instrumental in providing groundwork for subsequent macroscale experiments, namely the consolidation of calcium carbonate (CaCO3) plug using polyacrylamide (PAM) and to investigate the geomechanical strength of the consolidated calcium carbonate samples in the form of unconfined compressive stress (UCS) analysis. Results obtained from the geomechanical experiments have been compiled into a manuscript and currently under review to be published in another journal. Refer line 606 – 610.

Reviewer 2 Report

This is an interesting application of AFM force spectroscopy! This paper has the merit to be published in this journal after major review. 

(1) Were these measurements done in liquid or air? If air, do authors believe they resemble the accuracy of real life applications? If liquid, they need to more clearly specify that in the paper. 

(2) They mention there could be accumulation on the tip, could they comment on how that is not affecting the separation force measured? 

(3) What was the stiffness of the cantilever used? What was the method for finding it? This is an important value while you are measuring forces in nN. 

(4) Would you think the approach velocity on these molecules could play a role? Meaning if you approach them with higher or lower velocity, you could get completely different results? 

(5) during force spectroscopy, it is not ONLY repulsive or ONLY attractive. It is always a combination of both, however, the NET force can be either of them. That needs to be addressed in the paper. 

(6) Some editing is needed for the paper: 

(i) any abbreviation should have its full term shown first. 

(ii) some of the graph's labels are not clear. 

(iii) Sentence on line 81 in Page 2 could be improved. 

(iv) ....

This paper requires to be proof read and edited. 

Author Response

We wish to thank the reviewer for having read our manuscript and for the positive comments and suggestions to improve it. We know how much time it takes to review manuscripts and we sincerely appreciate the observations. We have acted on all of the remarks, and we feel they have improved significantly the manuscript.

Below we make a point-by-point discussion. The original reviewer’s comments are placed in italic black the response is red below each point.

Reviewer 2

(1) Were these measurements done in liquid or air? If air, do authors believe they resemble the accuracy of real life applications? If liquid, they need to more clearly specify that in the paper. 

Author reply:

All the force mapping experiments were conducted in liquid mode, where the AFM tip will be immersed into the dilute polymer solutions and run scan on the calcite surface immersed in polymer solutions. Explanation has been added in line 202 – 205.

(2) They mention there could be accumulation on the tip, could they comment on how that is not affecting the separation force measured? 

Author reply:

It is important to note that since each experiment fresh calcite surface and new AFM tip modified with CaCO3 was used, thus we anticipated a likelihood of the CaCO3 on the tip peeling away some of the HPAM molecules away from the calcite surface during the first few pixels, when the CaCO3 on the tip is relatively unoccupied. However, as the scans continued the CaCO3 surface on the AFM tip will slowly be saturated. We see this as the system achieving equilibrium, and thereafter we should observe the mutual detachment of HPAM molecules from both surfaces, that is HPAM detached from the calcite surface onto CaCO3 on AFM tip, and the HPAM detached from the CaCO3 on the tip onto the calcite surface. We believe by having a significantly large scan size of more than 1000 scans per sample, we are able to equilibrate the system so that ultimately, we are observing the net interaction of HPAM and CaCO3, regardless of the direction of detachment. Explanation has been added in line 219 – 228.

(3) What was the stiffness of the cantilever used? What was the method for finding it? This is an important value while you are measuring forces in nN.  

Author reply:

The procedure of obtaining the stiffness of the cantilever is explained in line 205 – 211. First, the AFM tip was made to contact a hard surface in the liquid, and a force curve is generated. The slope of the approach curve is selected, in which this tells us the sensitivity of the cantilever with respect to the movement of piezo. Then the thermal noise measurement was conducted with correction factor of 0.251. Hence, we were able to obtain the stiffness of the cantilever, which was approximately 0.3N/m for all the cantilevers used.

(4) Would you think the approach velocity on these molecules could play a role? Meaning if you approach them with higher or lower velocity, you could get completely different results? 

Author reply:

We have not conducted experiments at different approach velocity, as we do not see how this parameter can directly benefit our investigation on the effect of polymer molecular weight on the interaction with CaCO3. The approach velocity of the tip was set to be 2 µm/s, as this speed gave us reasonable experiment time without observing the hydrodynamic effect in the case of too high of an approach velocity. Explanation is added in line 214 – 216.

(5) during force spectroscopy, it is not ONLY repulsive or ONLY attractive. It is always a combination of both, however, the NET force can be either of them. That needs to be addressed in the paper. 

Author reply:

We fully agree with the reviewer that the force curves observed in our work is the net force of repulsive and attractive force. We sincerely appreciate this reminder from the reviewer and thus we have added an explanation in line 344 – 345.

(6) Some editing is needed for the paper: 

(i) any abbreviation should have its full term shown first. 

Author reply: we have gone through the manuscripts, and we believe we have added the full term of any abbreviation used.  

(ii) some of the graph's labels are not clear. 

Author reply: we have gone through the manuscripts, and we believe the graph’s labels which are less clear are that of figure 3 and 4. Thus we have colour coded the approach and retraction curves of these two figures, and further enlarged in the inset in figure 4.

(iii) Sentence on line 81 in Page 2 could be improved. 

Author reply: the sentence structure of line 80 - 82 have been corrected to clarify the meaning of the sentence.

Comments on the Quality of English Language: This paper requires to be proofread and edited. 

Author reply: the authors have read through the manuscript and made necessary correction on any language deficiency. We are happy to use the proofread service by the journal.

Round 2

Reviewer 1 Report

No comments

Reviewer 2 Report

All comments are addressed.